# Challenges and Future Trends in the Treatment of Psoriasis

**DOI:** 10.3390/ijms241713313

**Published:** 2023-08-28

**Authors:** Hyun-Ji Lee, Miri Kim

**Affiliations:** Department of Dermatology, Yeouido St. Mary’s Hospital, College of Medicine, The Catholic University of Korea, Yeongdeungpo-gu, Seoul 07345, Republic of Korea; o0or5r5r@gmail.com

**Keywords:** psoriasis, treatment, future trends

## Abstract

Psoriasis is a chronic inflammatory skin disorder, and current treatments include topical therapies, phototherapy, systemic immune modulators, and biologics, aiming to alleviate symptoms and improve quality of life. However, challenges persist, such as adverse effects, treatment resistance, high costs, and variability in response among individuals. The future of psoriasis treatment shows promising emerging trends. New biologic agents targeting novel pathways, such as interleukin 23 inhibitors like mirikizumab, offer enhanced efficacy. Small molecule inhibitors like RORγt inhibitors and ROCK2 inhibitors provide additional treatment options. Combination therapies, including biologics with methotrexate, may improve treatment response. Advancements in topical treatments utilizing microneedles and nanoparticle-based carriers can enhance drug delivery and improve therapeutic outcomes. Biomarkers and multi-omics technologies hold potential for personalized treatment approaches, thus aiding in diagnosis, predicting treatment response, and guiding therapeutic decisions. Collaboration among researchers, clinicians, and industry stakeholders is crucial to translating these scientific breakthroughs into clinical practice. By addressing current challenges and exploring these promising trends, we can optimize psoriasis management and improve the lives of those affected by this chronic condition.

## 1. Introduction

Psoriasis is a chronic immune-mediated inflammatory skin disorder that affects approximately 2–3% of the global population [1]. It is characterized by red, scaly plaques that can appear on various parts of the body, causing significant physical and psychological burden to affected individuals [2]. The pathogenesis of psoriasis involves complex interactions among genetic, immunological, and environmental factors, leading to dysregulated immune responses and excessive proliferation of keratinocytes.

Over the years, significant progress has been made in understanding the underlying mechanisms of psoriasis, leading to the development of various treatment options. Current management strategies for psoriasis aim to alleviate symptoms, improve quality of life, and prevent disease progression [3,4]. These approaches encompass topical therapies, phototherapy, systemic medications, and biologic agents targeting specific immune pathways. Despite the advancements in psoriasis management, several challenges and unmet needs persist. Adverse effects, treatment resistance, long-term safety concerns, and high costs associated with some therapies limit their widespread use and efficacy. Additionally, the variability in treatment response among individuals highlights the need for personalized and tailored approaches in psoriasis management [5,6].

Emerging trends and future directions in psoriasis treatment hold promise for improved outcomes. These include the development of novel biologic agents targeting novel pathways, the exploration of combination therapies to enhance efficacy and minimize side effects [7,8], the utilization of biomarkers for treatment selection and monitoring [9], and the advancement of gene- and cell-based therapies [10].

This review aims to provide an overview of the current treatment landscape for psoriasis, highlighting the challenges faced by existing therapies and discussing the potential future trends that may shape the field. By addressing these aspects, we hope to contribute to the ongoing efforts to optimize psoriasis management and improve the lives of individuals affected by this chronic condition.

## 2. Current Treatment Guidelines

### 2.1. Treatments for Mild Psoriasis

There is no consensus on the definitions of mild and moderate-to-severe psoriasis [11,12]. Mild psoriasis is generally described as affecting less than 3% to 5% of the total body surface area. There are several treatment options available for mild psoriasis, including topical corticosteroids, vitamin D analogs, calcineurin inhibitors, keratolytics, and targeted phototherapy [13,14,15]. The choice of treatment depends on factors such as the location and severity of the lesions, presence of comorbidities, and individual patient preferences.

Topical corticosteroids are commonly used as the primary therapy for patients with mild or localized psoriasis. They work by reducing inflammation, inhibiting cell proliferation, and constricting blood vessels through the downregulation of inflammatory pathways. The selection of corticosteroid strength and formulation should be based on the location of the lesions to minimize adverse effects. Combined formulations of corticosteroids with vitamin D analogs or keratolytic agents, such as halobetasol propionate and tazarotene, are often more effective and have fewer side effects compared to using them individually [16,17]. Additionally, they can also be used as proactive treatments applied twice a week when lesions show improvement [18].

Topical vitamin D analogs function by inhibiting the proliferation of keratinocytes and promoting their differentiation. They can be applied liberally unless the patient has renal impairment. Adverse effects may include a burning sensation and irritation, but these usually decrease over time [19]. Topical calcineurin inhibitors like tacrolimus and pimecrolimus are used primarily for psoriatic lesions in facial and intertriginous areas by blocking T cell activation and inhibiting the synthesis of IL-2 and IFN-γ. The main side effects of topical calcineurin inhibitors, similar to topical vitamin D analogs, are a burning sensation and skin irritation. These side effects can be more pronounced in areas with severe inflammation, and applying topical corticosteroids first can help reduce the likelihood of these side effects [20]. Topical keratolytics, such as tazarotene [17] and salicylic acid, aid in the breakdown of thick scales on psoriasis plaques. Tazarotene, a retinoid, inhibits keratinocyte proliferation, while salicylic acid reduces scaling. Adjusting the concentration, formulation, or frequency of application or combining them with topical corticosteroids can help minimize adverse effects such as burning and irritation. Last, targeted phototherapy, such as excimer light therapy, utilizes specific wavelengths of light to treat localized plaque psoriasis. It has a low potential for carcinogenicity and can lead to significant improvement after approximately two months of treatment. Adverse effects may include a burning sensation and blistering, which are preventable with an appropriate treatment schedule [21].

### 2.2. Treatments for Moderate-to-Severe Psoriasis

Moderate psoriasis is usually defined as psoriasis affecting from 3~5% to 10% of the body surface area. Severe psoriasis is typically characterized by a body surface area coverage of 10% or more. Systemic treatments are the primary approach for moderate-to-severe psoriasis, and these can also be used for localized disease or when topical therapies are ineffective. Both the U.S. and European guidelines recommend biologics, oral agents, and phototherapy in combination for these patients [22,23,24,25]. Biologics have shown higher efficacy compared to oral medications or phototherapy [26]. Topical therapies can be used as supplementary treatments but not as standalone therapy for moderate-to-severe psoriasis.

#### 2.2.1. Phototherapy

Phototherapy, including narrowband UV-B, broadband UV-B, and PUVA, has been used to treat moderate-to-severe psoriasis. Narrowband UV-B is preferred over the broadband form due to its higher effectiveness and better safety profile. UV-B phototherapy reduces DNA synthesis, leading to apoptosis of keratinocytes and decreased production of pro-inflammatory cytokines. Adverse effects may include erythema, pruritus, blistering, photoaging, and photocarcinogenesis. Narrowband UV-B is more commonly used due to its greater efficacy, longer remission duration, lower potential for skin cancer, and reduced erythema compared to broadband UV-B. Combining narrowband UV-B with systemic retinoids may enhance efficacy and reduce the potential for skin cancer [24]. PUVA therapy involves the use of psoralens, such as methoxalen, to suppress DNA synthesis followed by UV-A irradiation. Although oral PUVA is more effective than UV-B, it is no longer preferred due to the increased risk of skin cancer with long-term use. Adverse effects may include gastrointestinal upset, burning, pruritus, hypertrichosis, and photoaging. Topical PUVA therapy is commonly used for palmoplantar psoriasis, involving soaking hands and feet in water with psoralen followed by UV-A irradiation. The main challenge with phototherapy is the need for patients to travel to undergo office-based sessions. Home phototherapy is a convenient option but may be limited by insurance and space constraints.

#### 2.2.2. Oral Systemic Treatments

Before the introduction of biologics, oral agents were commonly used to treat moderate-to-severe plaque psoriasis. The available oral treatment options for plaque psoriasis include methotrexate, apremilast, acitretin, and cyclosporine [23]. Compared to biologics, the efficacy of oral treatments is generally low, except for that of cyclosporine. However, oral medications may still be considered for patients who have limited access to biologics or prefer non-injectable treatments. The adverse effect profiles differ significantly among the oral options, and careful consideration is necessary when selecting an oral agent due to various contraindications and precautions associated with them.

#### 2.2.3. Biologics for the Treatment of Moderate-to-Severe Plaque Psoriasis

TNF-α inhibitors

TNF-α inhibitors are a class of medications that target tumor necrosis factor-alpha (TNF-α), a cytokine involved in inflammation. Three commonly used TNF-α inhibitors are etanercept, infliximab, and adalimumab. The response to TNF-α inhibitors is typically observed after 12 to 16 weeks of continuous treatment, except for infliximab, where response is experienced after 8 to 10 weeks. Their efficacies and long-term safety profiles have been demonstrated in moderate-to-severe psoriasis [27,28]. However, many severe adverse events were reported, such as serious infections, reactivation of hepatitis B and C, tuberculosis, drug-induced lupus, and demyelinating central nervous system disorders [29,30]. TNF-α inhibitors may be beneficial for patients with a history of inflammatory bowel disease, and certain inhibitors are approved for its treatment.

IL-23 inhibitors

Ustekinumab, guselkumab, risankizumab, and tildrakizumab are effective IL23 inhibitors used in the treatment of psoriasis. Mirikizumab is for use in late-phase development. Ustekinumab is the only biologic that targets both IL-12 and IL-23 by inhibiting their shared p40 subunit. In clinical studies, ustekinumab at doses of 45 mg and 90 mg showed PASI 75 response rates of 67.5% and 73.8%, respectively, at week 12. Other IL-23 inhibitors also demonstrate robust efficacy, acceptable safety profiles, and convenient dosing regimens. Guselkumab at 100 mg demonstrated PASI 75/90/100 response rates of 91.2%/73.3%/37.4%, respectively, at week 16, while risankizumab at 150 mg showed rates of 90.8%/74.8%/50.7%. Tildrakizumab at 100 mg exhibited respective PASI 75/90/100 response rates of 77%/54%/23% at week 28 [22]. Safety profiles are similarly acceptable, with no increased risk of serious infections or malignancies. Common side effects included nasopharyngitis, upper respiratory tract infection, headache, and fatigue.

IL-17 inhibitors

IL-17 inhibitors target either the IL-17 ligand or its receptor. Secukinumab and ixekizumab inhibit IL-17A, while bimekizumab inhibits both IL-17A and IL-17F. Brodalumab targets IL-17 receptor α. IL-17 inhibitors have a rapid onset of action, strong response, and sustainable efficacy for plaque psoriasis. Secukinumab at 300 mg demonstrated PASI 75/90/100 response rates of 77.1%/54%/24%, respectively, at week 16, while ixekizumab at 80 mg after an initial 160 mg showed rates of 90%/70%/40%, and brodalumab at 210 mg showed rates of 83%/70%/42% [31,32,33]. They are also approved for psoriatic arthritis. In addition, secukinumab and ixekizumab have been reported to be particularly effective in treating nail psoriasis. The safety profile of IL-17 inhibitors is acceptable, with no increased risk of serious infections or malignancy. However, mucocutaneous candidiasis and exacerbation of inflammatory bowel disease have been reported [34,35]. Common adverse reactions include upper respiratory tract infections and injection site reactions. A case of suicidal ideation has been reported in a patient treated with brodalumab, and the relationship between the two has not yet been clarified [36].

## 3. Challenges in the Current Treatment of Psoriasis

Biologic agents have revolutionized the treatment landscape for psoriasis, particularly in patients with moderate-to-severe disease. However, despite their effectiveness, several challenges and limitations persist in the use of these medications. One of the major challenges is the occurrence of primary and secondary treatment failures or inadequate responses to initial treatment. While biologic agents have shown high response rates, there is a subset of patients who do not respond to these treatments, leading to primary treatment failure [37]. This highlights the need for better predictors of treatment response to identify those patients who are more likely to benefit from a particular biologic agent. To address this challenge, the Psoriasis Stratification to Optimise Relevant Therapy (PSORT) consortium has recently convened to identify and study factors that contribute to treatment non-response. The details will be analyzed further in the following sections.

Secondary failure is defined as decreased efficacy in a patient who initially responded well to a treatment. One possible reason for the decrease in effectiveness is the development of neutralizing antibodies in patients. In other immune-mediated diseases, combining biologic agents with methotrexate has been shown to improve the longevity of biologic drug response [38]. However, there is limited reported evidence on the use of methotrexate, a conventional systemic therapy, in combination with biologics specifically for psoriasis treatment, and there are currently no randomized controlled trials (RCTs) to support a recommendation for combination therapy [39]. Similarly, switching biologics may also be a good option for secondary failure. To date, several studies on biologic change for psoriasis treatment have been published. The most frequent reason for discontinuing initial treatment was inefficacy. Patients who switched to a different biologic agent demonstrated comparable improvements in PASI scores to biologic-naive patients. Transitioning to a second biologic treatment due to inefficacy or adverse events associated with the initial one has the potential to enhance psoriasis management [40,41,42,43,44]. One study examined the real-world switching patterns in patients with psoriasis who were treated with biologic medications in the United States. They found that the overall rates of switching were 14.4% at 12 months and 26.0% at 24 months. IL-23 inhibitors were associated with the lowest risk of switching compared to TNF, IL-17, and IL-12/23 inhibitors. Factors such as prior use of targeted immune modulators, age, and female gender were identified as predictors of switching [43]. Considering a switch to a different biologic agent, the optimal interval between discontinuation of the previous medication and initiation of a new biologic is not well established. While the half-life of biologic agents can serve as a reference, individualized approaches are often required based on the patient’s disease severity, response to prior treatment, and expert opinion.

Another significant concern associated with biologic therapies is their high cost. These medications are often expensive, placing a financial burden on patients and healthcare systems. In a survey of patients with psoriasis, the majority of patients answered that they needed a more cost-effective option in treatment [45]. However, as the patents for some biologic agents approach expiration, the development of biosimilars offers the potential for reduced drug costs. Their availability can enhance cost-effectiveness, improve accessibility, and provide additional treatment options for patients. Biosimilars to the TNF-α inhibitor showed similar effects to the originator [46,47]. Accordingly, biosimilars to the IL-17 inhibitor and IL-23 inhibitor are expected to produce similar effects.

While the focus of biologic therapies has been predominantly on moderate-to-severe psoriasis, the treatment of mild psoriasis continues to rely heavily on localized therapies such as topical corticosteroids, vitamin D analogues, and calcineurin inhibitors. While these treatments can effectively control symptoms in many cases, there is a need for more targeted and effective systemic therapies for patients with mild psoriasis who do not respond adequately to topical treatments.

Additionally, certain subtypes of psoriasis, such as scalp [48], nail [49], and genital [50] psoriasis, palmoplantar psoriasis (PPP) [51], and generalized pustular psoriasis (GPP) [52] pose unique challenges in terms of treatment satisfaction and management. Current treatment options for these specific subtypes often fall short in providing satisfactory results. There is a need for further research and development to identify more effective and tailored approaches to address the specific needs of these patient populations.

## 4. Future Aspects in the Treatment of Psoriasis

As discussed in previous sections, while significant progress has been made in the management of psoriasis, current treatment options have several limitations. However, ongoing research and clinical trials have uncovered promising agents and innovative therapeutic approaches that hold the potential to revolutionize psoriasis treatment. This discussion aims to explore the future aspects of psoriasis treatment, including novel agents under development, advancements in topical therapies, personalized approaches, biomarker development, and the application of multi-omics technologies.

### 4.1. Biologic Agents and Small Molecule Inhibitors under Development

In recent years, the development of biologic agents and small molecule inhibitors targeting specific pathways implicated in psoriasis pathogenesis has garnered significant attention. Figure 1 describes the pathogenesis of psoriasis.

#### 4.1.1. IL-23 Inhibitors

Mirikizumab, an interleukin (IL)-23 inhibitor, has shown promising results in clinical trials. By targeting the IL-23/Th17 pathway, mirkikizumab effectively suppresses inflammation and leads to significant improvements in psoriatic skin lesions [53]. In psoriasis pathophysiology, the IL-23/IL-17 axis plays a crucial role, and currently available biologics are designed to target this molecule.

#### 4.1.2. JAK Inhibitors

Type 1 and type 2 cytokine receptors rely heavily on the Janus kinase (JAK) and signal transducer and activator of transcription (STAT) pathways. The JAK family consists of intracellular protein tyrosine kinases, including JAK1, JAK2, JAK3, and tyrosine kinase 2 (TYK2), while the STAT family comprises STAT1, STAT2, STAT3, STAT4, STAT5 (STAT5A and STAT5B), and STAT6. In psoriasis, the crucial IL-23 receptor is associated with JAK2, TYK2, and STAT3 [54]. While JAK 1–3 inhibitors have demonstrated efficacy in moderate-to-severe psoriasis patients, safety concerns have persisted, and no JAK inhibitor has received regulatory approval for the treatment of psoriasis [55]. Meanwhile, deucravacitinib is a selective TYK2 inhibitor that targets key inflammatory pathways involved in psoriasis pathogenesis. Clinical trials have demonstrated its effectiveness in reducing psoriasis symptoms and improving patients’ quality of life [56].

#### 4.1.3. IL-36 Inhibitors

IL-36, a member of the IL-1 family, binds to its receptor and activates the NF-κB and MAPKs pathways via the MyD88/IRAK complex [57]. A severe form of generalized pustular psoriasis (GPP) has been associated with loss-of-function mutations in the IL-36Ra gene [58], as has plaque psoriasis. Recently, a small molecule inhibitor of IL-36 called A-552 has been shown to effectively inhibit IL-36γ and the production of other cytokines induced by IL-36γ in both human and mouse cells [59]. Furthermore, a monoclonal antibody targeting the IL-36 receptor, spesolimab, has demonstrated efficacy in a Phase I study, and Phase II and III studies evaluating its use in GPP are currently underway [60]. Therefore, anti-IL-36 agents hold significant potential in the treatment of psoriasis, and further research is needed to evaluate their efficacy and safety.

Additionally, recent studies have suggested the potential therapeutic role of IL-37 and IL-38, which are MAST cell cytokines, in the treatment of psoriasis [61].

#### 4.1.4. RORγt Inhibitors

RORγt inhibitors are a class of drugs that specifically target retinoic-acid-receptor-related orphan receptor gamma-t (RORγt), a transcription factor essential for the development and function of Th17 cells. By modulating the Th17-cell-mediated immune response implicated in psoriasis, RORrt inhibitors show potential as a novel therapeutic approach [62]. VTP-43742, an oral inhibitor of the RORγT protein, is currently being evaluated in a phase III clinical trial for the treatment of plaque psoriasis. In a phase IIa study, patients receiving 700 mg and 350 mg of VTP-43742 demonstrated reductions of 29% and 23%, respectively, in the PASI score after four weeks. Additionally, both treatment groups exhibited a significant decrease in levels of IL-17A and IL-17F, cytokines associated with psoriasis [63]. Adverse effects reported in the study included headache, flushing, elevated liver enzymes, and nausea.

#### 4.1.5. ROCK2 Inhibitors

Rho-associated protein kinase 2 (ROCK2) inhibitors have emerged as potential candidates for psoriasis treatment. By inhibiting ROCK2, these agents modulate inflammatory responses and reduce the production of pro-inflammatory cytokines, offering a new avenue for targeted therapy. One study demonstrated that oral administration of a specific ROCK2 inhibitor, KD025, lead to a 50% reduction in PASI scores from the baseline in 46% of patients with psoriasis vulgaris and significant reductions in IL-17 and IL-23 levels, while IL-6 and TNF-α levels remained unchanged. These results indicate that an orally available selective ROCK2 inhibitor effectively downregulates the autoimmune response driven by Th17 cells, leading to improved clinical symptoms in psoriatic patients [64].

### 4.2. Advancements in Topical Treatments

Over the past decades, there has been a growing number of studies on psoriasis, and researchers have explored various biomaterials to enhance the effectiveness of topical anti-psoriatic drug delivery for better treatment outcomes of psoriatic plaques [65]. The representative biomaterials used include nanoparticles, microneedles, nanofibers, and hydrogels. Among these, nanoparticles have been the most widely studied, with a significant increase in research publications in recent years. Microneedles and nanofibers are also emerging as promising areas in psoriasis research. Hydrogels have long been utilized as carriers for anti-psoriatic agents.

#### 4.2.1. Microneedles

Over the past decades, microneedles (MNs) have gained popularity for transdermal drug delivery [66]. They consist of a baseplate with an array of microsized needles, capable of piercing the skin and creating microchannels. Different types of MNs exist, including solid microneedles (SMNs), coated microneedles (CMNs), dissolving microneedles (DMNs), and hollow microneedles (HMNs) [67].

In recent years, MNs have shown promise in the treatment of psoriasis. Clinical trials and animal studies have demonstrated their effectiveness in enhancing drug permeation and improving therapeutic outcomes. One study conducted a small clinical trial on psoriatic patients resistant to topical calcipotriol/betamethasone ointment therapy. They used an SMN patch based on hyaluronic acid (HA), which was placed over the lesions after ointment application, resulting in improved therapeutic effects with the assistance of MNs [68]. In another study using an imiquimod-induced psoriasiform model in mice, HA-based DMNs with methotrexate encapsulated in the needles were applied. These DMNs successfully penetrated the skin and released the drug, leading to amelioration of psoriatic lesions with lower drug doses and fewer systemic side effects compared to oral methotrexate administration [69]. More recently, researchers developed an HA-based DMN patch to mitigate inflammatory skin diseases, including psoriasis. The DMNs were used for transdermal delivery of NLRP3-targeted Cas9 nanocomplexes and nanoparticles loaded with dexamethasone (Dex), which specifically inhibited NLRP3 inflammasome via CRISPR-Cas9-induced genome editing, further enhancing psoriasis therapy [70]. These studies demonstrate the potential of microneedles as a novel and effective approach for psoriasis treatment, offering targeted drug delivery and reduced side effects.

#### 4.2.2. Topical Nanocarriers

Nanoparticles (NPs) have been studied extensively for their potential in transdermal drug delivery, particularly in the treatment of psoriasis. NPs offer advantages such as a large surface-area-to-volume ratio, enabling them to penetrate through hair follicles and the skin’s various routes, including the intercellular lipid route, transcellular route, and follicular route [71]. The characteristics of NPs, such as size, molecular weight, surface charge, and pH, influence their drug permeation through the skin. For psoriasis treatment, adjusting the characteristics of NPs is crucial to enhance their permeation through the altered skin conditions.

In psoriasis management, lipid-based NPs have been widely investigated. These include solid lipid nanoparticles (SLNs), nanostructured lipid carriers (NLCs), liposomes, transfersomes, ethosomes, and niosomes. Lipid-based NPs are advantageous for transdermal drug delivery due to their flexibility, which allows them to pass through the extracellular matrix in the skin’s stratum corneum layer. For instance, NLCs loaded with anti-psoriatic drugs like methotrexate have shown promising results in enhancing drug permeation and therapeutic effects [72]. Metallic nanoparticles, such as gold and silver nanoparticles, have also been explored for drug delivery in psoriasis. These rigid nanocarriers offer tunable size and surface characteristics. Functionalized gold nanoparticles, for example, have demonstrated improved drug delivery and immunomodulatory effects in psoriatic skin [73,74,75]. Polymeric nanoparticles, including chitosan and poly(lactide-co-glycolic acid) (PLGA) nanoparticles, have also been studied extensively. Synthetic polymeric NPs offer defined characteristics for drug encapsulation and release. PLGA NPs loaded with curcumin have shown enhanced drug permeation through psoriatic skin and improved therapeutic effects [76]. Nanofibers, fabricated through electrospinning and other methods, have emerged as potential vehicles for drug delivery in psoriasis treatment. These nanofiber-based delivery systems have been used to load anti-psoriatic agents for controlled release and enhanced treatment efficacy [77,78].

In conclusion, NPs and nanofibers hold great promise for transdermal drug delivery in psoriasis management. However, further research is needed to fully understand the specific penetration and drug release mechanisms of different types of NPs and to develop targeted NPs for improved therapeutic efficiency and reduced side effects in psoriatic skin.

### 4.3. Biomarker Development and Application of Multi-Omics

Identification and validation of reliable biomarkers are crucial for optimizing psoriasis treatment. Biomarkers can aid in disease diagnosis, predict treatment response, and guide therapeutic decisions. Ongoing research focuses on identifying biomarkers associated with disease activity, therapeutic response, and the development of comorbidities, enabling more personalized and effective treatment strategies [9]. A recent review described potential biomarkers of disease progression in psoriasis. The study identified several potential candidate biomarkers that show promise for predicting disease severity in psoriasis. These include LCE3D, IL23R, IL23A, NFKBIL1 loci, and HLA-C*06:02 at the genomic level. In the proteomic realm, candidates such as IL-17A, IgG aHDL, GlycA, I-FABP, and kallikrein 8 were highlighted. Additionally, metabolomic candidates like tyramine were also considered. Regarding the prediction of type 2 diabetes mellitus, the study pointed out genomic variations in the IL12B and IL23R loci as potential indicators. However, it is important to note that none of these biomarkers had sufficient evidence to be immediately employed for clinical decision making. Further validation is required to establish their clinical utility effectively [79].

The integration of multi-omics technologies, including genomics, transcriptomics, proteomics, and metabolomics, holds great potential for unraveling the complex molecular mechanisms underlying psoriasis. This multi-dimensional approach may lead to the development of more effective therapies and personalized treatment strategies.

## 5. Conclusions

In conclusions, psoriasis is a complex skin disorder with significant physical and emotional impacts on affected individuals. Despite the progress in understanding its underlying mechanisms and the development of various treatment options, challenges remain in achieving optimal outcomes. The future of psoriasis treatment looks promising with the emergence of novel biologic agents targeting specific pathways, such as IL-23 inhibitors like mirikizumab and RORγt inhibitors. These agents offer the potential for enhanced efficacy and better disease control. Advancements in topical treatments, particularly with microneedles and nanoparticle-based carriers, hold promise for improving drug delivery and treatment effectiveness for psoriatic plaques. These innovative approaches may provide more targeted and efficient therapies, minimizing side effects and improving patient adherence. The identification and validation of biomarkers using multi-omics technologies are crucial steps toward personalized treatment approaches. Biomarkers can help predict treatment response, monitor disease activity, and guide therapeutic decisions, ultimately leading to more individualized and effective treatment strategies. However, while these advancements show great potential, collaboration among researchers, clinicians, and industry stakeholders is vital to translate these discoveries into clinical practice. Additionally, addressing the challenges of treatment resistance, adverse effects, and high costs remains essential in optimizing psoriasis management. By embracing emerging trends, fostering collaboration, and tailoring treatments to individual needs, we can continue to make significant strides in the field of psoriasis management. Ultimately, this will lead to improved outcomes and a better quality of life for individuals living with this chronic skin condition.

## Figures and Tables

**Figure 1 ijms-24-13313-f001:**
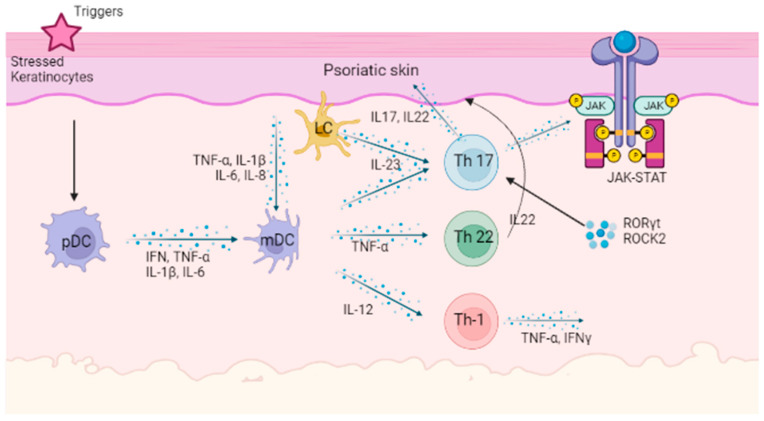
Pathophysiology of psoriasis. Keratinocytes can be prompted by initial triggers, and when under stress, these keratinocytes release self-nucleotides and antimicrobial peptides. This activates pDCs followed by mDCs, playing a crucial role in the initial phase of psoriasis initiation. Stimulated DCs release pro-inflammatory factors (IL-12, IL-23, and TNF-α). These cytokines in turn induce Th17, Th22, and/or Th1 cell differentiation. Furthermore, activated Th17 cells promote TYK2/STAT3 signals in keratinocytes. These promote inflammatory infiltration, epidermal hyperplasia, innate immunity, and tissue reorganization. pDC: plamacytoid dendritic cell; mDC: myeloid dendritic cells; IFN: interferon; TNF-α: tumor necrosis factor-α; IL: interleukin; LC: Langerhans cell; Th: T helper cell; RORγt: retinoic-acid-receptor-related orphan receptor gamma-t; ROCK2: Rho-associated protein kinase 2; TYK2: tyrosine kinase 2; JAK: Janus kinase; STAT: signal transducer and activator of transcription.

## Data Availability

Not applicable.

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
