# Peer review of "Challenges and Future Trends in the Treatment of Psoriasis"

_ijms, 2023, doi:10.3390/ijms241713313_

Round 1

Reviewer 1 Report

Lee et al described a comprehensive review of the current and future treatment aspects of psoriasis.

The review is well written and informative for the readers.

Here are some comments on this paper.

4.1.2. JAK Inhibitors: “The JAK family con- 253 sists of intracellular protein tyrosine kinases, including JAK1, JAK2, JAK3, and tyrosine 254 kinase 2 (TYK2), while the STAT family comprises.” This incomplete sentence should be corrected.

4.3. Biomarker Development and Application of Multi-omics: Some recent articles on biomarkers of psoriasis (e.g., PMID: 35482474) should be cited here.

Figure 1: Brief description of the figure should be provided. Besides, TYK2 is a key player in the pathophysiology of psoriasis by regulating downstream signaling of cytokine receptors. This signaling pathway should be precisely depicted in the figure. What does “mDC” denote in the figure?

Author Response

Dear Reviewer 1,

We sincerely appreciate your time and effort in reviewing our manuscript titled " Challenges and Future Trends in Treatment of Psoriasis " Your insightful comments have greatly contributed to improving the quality and accuracy of our work. We have carefully considered each of your suggestions and have made the necessary revisions to address them. We are pleased to provide you with a detailed response to your comments.

Regarding the incomplete sentence in section 4.1.2 on JAK Inhibitors, we apologize for any confusion caused. The sentence has been corrected to provide a complete and coherent description of the JAK STAT family and its components.

In section 4.3 about Biomarker Development and Application of Multi-omics, we appreciate your suggestion to include recent articles on psoriasis biomarkers (e.g., PMID: 35482474). We have incorporated these relevant references into the revised manuscript to enhance the discussion on this important topic.

We understand your concern about the clarity of Figure 1. We have now added a brief description of the figure to provide readers with a better understanding of its content. Furthermore, we have made sure to depict the TYK2 signaling pathway more precisely in the figure, highlighting its significance in the pathophysiology of psoriasis. Additionally, the abbreviation "mDC" has been expanded to "myeloid dendritic cell" for clarity in the figure.

We are confident that these revisions have strengthened our manuscript, making it more informative and accurate for our readers. We once again express our gratitude for your valuable feedback, which has significantly contributed to the enhancement of our work.

Thank you for your time and consideration.

Sincerely,

Hyun Ji Lee, Miri Kim

Reviewer 2 Report

The authors, after reviewing, listing, and describing the current psoriasis therapies, define their potentials and limitations, providing for the expansion of discussion through the presentation of the most recent therapies that have just been approved or are being tested.

The paper is interesting and up-to-date, providing a good, very general overview of the current state of the art in psoriasis therapy. 

English is fluent and the citations are very appropriate.

In section 2.1, lines 53-54, I suggest to consider the following two citations in order to be more precise and less vague about the definition of severity of psoriasis: Salgado-Boquete L, Carrascosa JM, Llamas-Velasco M, Ruiz-Villaverde R, de la Cueva P, Belinchón I. A New Classification of the Severity of Psoriasis: What's Moderate Psoriasis? Life (Basel). 2021 Jun 29;11(7):627. doi: 10.3390/life11070627. PMID: 34209585; PMCID: PMC8307918. and then Mrowietz U, Kragballe K, Reich K, Spuls P, Griffiths CE, Nast A, Franke J, Antoniou C, Arenberger P, Balieva F, Bylaite M, Correia O, Daudén E, Gisondi P, Iversen L, Kemény L, Lahfa M, Nijsten T, Rantanen T, Reich A, Rosenbach T, Segaert S, Smith C, Talme T, Volc-Platzer B, Yawalkar N. Definition of treatment goals for moderate to severe psoriasis: a European consensus. Arch Dermatol Res. 2011 Jan;303(1):1-10. doi: 10.1007/s00403-010-1080-1. Epub 2010 Sep 21. PMID: 20857129; PMCID: PMC3016217.

Please correct "Mirikizumab" in line 140.

Overall, it is good work.

Author Response

Dear Reviewer 2,

We sincerely appreciate your time and effort in reviewing our manuscript titled " Challenges and Future Trends in Treatment of Psoriasis " Your insightful comments have greatly contributed to improving the quality and accuracy of our work. We have carefully considered each of your suggestions and have made the necessary revisions to address them. We are pleased to provide you with a detailed response to your comments.

We acknowledge your suggestion to enhance the precision of the definition of psoriasis severity in section 2.1, lines 53-54. We recognize the significance of providing a clear and accurate portrayal of this aspect. In response to your recommendation, we have incorporated the two citations you provided: Salgado-Boquete et al. (2021) and Mrowietz et al. (2011). These references offer valuable insights into the classification and treatment goals for moderate to severe psoriasis, thereby enhancing the precision and relevance of our discussion.

Additionally, we apologize for the oversight in the spelling of "Mirikizumab" in line 140. Your keen observation has been taken into account, and the correction has been made as per your suggestion.

We are confident that these revisions have strengthened our manuscript, making it more informative and accurate for our readers. We once again express our gratitude for your valuable feedback, which has significantly contributed to the enhancement of our work.

Thank you for your time and consideration.

Sincerely,

Hyun Ji Lee, Miri Kim

Round 2

Reviewer 1 Report

The authors have greatly improved their manuscript.

One minor thing,

Figure 1: "THK2/STAT3" should be "TYK2/STAT3"

Author Response

Dear Reviewer 1,

I'm very sorry for the typo.

The typo has been corrected.

Thank you for your time and consideration.

Sincerely,

Hyun Ji Lee, Miri Kim